# Parvovirus B19 Outbreak in Israel: Retrospective Molecular Analysis from 2010 to 2023

**DOI:** 10.3390/v16030480

**Published:** 2024-03-20

**Authors:** Orna Mor, Marina Wax, Shoshana-Shani Arami, Maya Yitzhaki, Or Kriger, Oran Erster, Neta S. Zuckerman

**Affiliations:** 1Medical School, Tel-Aviv University, Tel Aviv 6997801, Israel; 2Central Virology Laboratory, Ministry of Health, Chaim Sheba Medical Center, Ramat Gan 5262112, Israel; marina.wax@sheba.health.gov.il (M.W.); shoshanaarami.arami@sheba.health.gov.il (S.-S.A.); mayan.yezchaki@sheba.health.gov.il (M.Y.); oran.erster@sheba.health.gov.il (O.E.); neta.zuckerman@sheba.health.gov.il (N.S.Z.); 3Clinical Microbiology and Pediatric Infectious Disease Unit, Chaim Sheba Medical Center, Ramat Gan 5262112, Israel; or.kriger@sheba.health.gov.il

**Keywords:** parvoB19, outbreak, RT-PCR, NGS

## Abstract

This study presents an analysis of the epidemiological trends of parvovirus B19 (B19V) in Israel from 2010 to 2023, with particular emphasis on the outbreak in 2023. The analysis utilized molecular diagnostic data from individual patients obtained at the Central Virology Laboratory. Between 2010 and 2022, 8.5% of PCR-tested samples were positive for B19V, whereas in 2023, this percentage surged to 31% of PCR-tested samples. Throughout the study period, annual cycles consistently peaked in early spring/summer, with the most recent prominent outbreak occurring in 2016. Predominantly, diagnoses were made in children and women aged 20–39. Despite the notable surge in 2023, over 80% of positive cases continued to be observed in children and young women, with a decrease in cases during winter months. Furthermore, genotype 1a of the virus remained the predominant strain circulating during the outbreak. In light of these circumstances, consideration should be given to implementing screening measures, particularly among high-risk groups such as pregnant women.

## 1. Introduction

Human Parvovirus B19 (B19V) belongs to the family of Parvoviridae, genus *Erythroparvovirus*. The viral genome consists of 5596 nucleotides encoding for several structural and non-structural proteins among which are the non-structural protein 1 (NS1) and the capsid viral protein 1 (VP1) and VP2. A phylogenetic analysis based mainly on a fragment spanning the NS1/VP1 region has shown that this virus is a widely distributed viral pathogen comprising three distinct genotypes [1], with genotype 1 being the predominant strain circulating globally [2].

The primary mode of B19V transmission is through respiratory droplets. Infection of the respiratory tract with B19V can be asymptomatic or lead to a spectrum of clinical presentations. In children, the infection may manifest as the characteristic “slapped cheek” rash, commonly known as the fifth disease [3]. Among individuals with hematological disorders, transient aplastic crisis is a potential complication [4]. In pregnant women, B19V infection can result in non-immune hydrops fetalis [5,6]. Additionally, persistent anemia [7], arthropathy, and inflammation of various tissues [8] may occur as a consequence of B19V infection.

B19V infections exhibit seasonal patterns, with regions experiencing moderate climates typically observing higher incidence rates from late winter to early summer [9]. Epidemics tend to occur at intervals of approximately 4–5 years [4]. 

In many cases, especially in children, B19V infections are clinically diagnosed by pediatricians without laboratory confirmation. However, when laboratory diagnosis is requested, it is typically performed by serology using commercial ELISA or immunofluorescence-based assays that detect anti B19V IgG and IgM antibodies. Despite PCR-based molecular detection being the preferred method for detecting respiratory viral infections, it is not widely employed for B19V diagnosis, and most routine clinical laboratories rely solely on serology. In Israel, molecular diagnosis is centralized and exclusively conducted by the Central Virology Laboratory (CVL) of the Ministry of Health [10].

In early 2023, pediatric physicians across Israel reported a notable rise in cases clinically diagnosed as the fifth disease. Simultaneously, the CVL and other clinical laboratories experienced a surge in requests for B19V diagnosis [11], although, still, most B19V cases remained clinically diagnosed without laboratory confirmation. Herein, we summarize the B19V molecular diagnoses at the CVL in 2010–2023 in light of the 2023 outbreak.

## 2. Materials and Methods

### 2.1. Clinical Samples

Molecular diagnosis of B19V is conducted using real-time PCR (RT-PCR) and is centralized within the CVL [12]. RT-PCR testing is typically requested in cases where B19V infection is suspected. For this analysis, the CVL database was screened for samples with RT-PCR results obtained between January 2010 and December 2023. Only one sample per patient (*n* = 2698) was included in the analysis. Sex and age at the time of sample collection were recorded. Clinical samples used for B19V testing encompassed whole blood or intrauterine samples, particularly in cases where diagnosis was requested for pregnant women.

### 2.2. Molecular Analysis

Molecular diagnosis of B19V was performed using a custom-made RT-PCR based on a published protocol [12]. For sequencing purposes, a 1.1 kb PCR fragment spanning the NS1/VP1u region of the virus was generated, utilizing the remains of the nucleic acid extracts used for the initial RT-PCR diagnosis and primers targeting nucleotides 1863–2953 of the NS1/VP1u B19V region. Three to four representative B19V-positive sample remains collected in each of the studied years were selected for sequencing (*n* = 44 overall). Sequences and data describing the sequenced samples are provided in Appendix A. Libraries were constructed using next-generation sequencing (NGS) technology following the Illumina DNA prep library protocol. Sequencing was performed on a MiSeq instrument (Illumina, San Diego, CA, USA).

### 2.3. Phylogenetic Analysis

Fastq files underwent quality control assessment using fastQC and MultiQC [13] and low-quality sequences were filtered using a trimmomatic [14]. Sequences were mapped to a B19V reference genome (NC_000883.2) using the Burrows–Wheeler aligner (BWA) mem [15]. BAM files were sorted and indexed using the SAMtools suite [16]. Coverage and depth of sequencing were calculated using the SAMtools suite [16]. Consensus fasta sequences were assembled for each sample using iVar [17], with positions with <5 nucleotides determined as Ns. Phylogenetic analysis was conducted using the Augur pipeline [18]. Multiple alignment of sequences with the B19V reference genome was performed using augur align, and a time-resolved phylogenetic tree was constructed with IQ-Tree [19] and TreeTime [20] under the GTR substitution model and visualized with auspice [18].

## 3. Results

### 3.1. B19V Positive Rates, 2010–2023

Between January 2010 and December 2023, the CVL received 3327 samples with requests for B19V RT-PCR-based molecular diagnosis. During that period, 9% (236/2698) of samples were B19V positive. The overall number of requests for PCR testing increased during the study period. From 2010 to 2015, the median number of requests was 67 (IQR 75), while from 2016 to 2022, the median number of unique samples transferred to the CVL for RT-PCR diagnosis increased to 315 (IQR 48). Prior to 2023, an increase in the rates of B19V positives was observed in 2011–2012 and in 2019–2020. However, a single major surge of B19V was recorded in 2016, with 16% of samples testing positive by PCR. In 2023, B19V molecular detection was requested for 629 samples, of which 31% tested positive for B19V DNA (Figure 1). 

The annual pattern of B19V diagnosis remained consistent from 2010 to 2022 and in 2023. The annual peak for positive B19V cases occurred consistently between the months of April and August, with the lowest diagnosis rates of B19V observed during the winter months, spanning from October to January (Figure 2).

### 3.2. Demographics of B19V Infections

Demographic analyses revealed a consistent pattern of B19V infections throughout the study period. The majority of B19V-positive cases, comprising 72% and 78% of all B19V cases detected during 2020–2022 and in 2023, respectively (*p* = 0.11), were found in individuals below the age of 39 (Figure 3a). Among individuals under 19 years old, there was no significant difference in the rate of infection between males and females. However, among those aged 20–39, a B19V-positive diagnosis was predominantly observed in women: 116 out of 131 (89%) in 2010–2022, and 59 out of 66 (89%) in 2023 (*p* = 0.86, Figure 3b). 

### 3.3. Phylogenetic Analysis of Circulating B19V

To characterize the molecular epidemiology of B19V in Israel and to compare the B19V sequences in 2023 to previous years, B19V samples collected between 2012 and 2023 were used for VP1 gene sequencing and phylogenetic analysis. In addition to the Israeli sequences, the phylogenetic analysis included global sequences (*n* = 246) from various B19V genotypes. The analysis shows that all Israeli sequences are clustered under genotype 1a (Figure 4). This cluster also contains sequences from other countries, mainly countries in Europe like The Netherlands, France, and Belarus and also a tight cluster of 1a sequences from China.

Israeli sequences, including those from the 2023 outbreak, did not exhibit geographic clustering (Figure 5a). However, sequences from the 2023 outbreak were observed to be grouped in a single cluster, characterized by three unique synonymous mutations (C2596T, G2680A, and C2794T, Figure 5b). Notably, this cluster also included sequences from patients diagnosed in previous years: one sequence from 2022, two sequences from 2020, and one sequence from 2013. 

## 4. Discussion

This study was initiated during the 2023 B19V outbreak with the aim of assessing the molecular characteristics of B19V circulating in 2023 and comparing it to B19V identified in previous years. Although the percentage of laboratory-confirmed positives was higher in 2023 compared to all previous years, the B19V sequences identified in 2023 belonged to the same 1a B19V genotype observed in previous years, which is the most commonly found B19V genotype worldwide. Unlike other viruses, sequencing of B19V is rarely performed and, therefore, global data on B19V transmissions are lacking. Nonetheless, our analysis reveals that B19V sequences from Israel exhibit clustering with other 1a sequences originating from neighboring European countries. This clustering phenomenon may be attributed to the close interactions among these countries, facilitated by factors such as membership in the European Union and ongoing connections with Israel. Notably, sequences from China classified as 1a formed a distinct cluster within this Israeli group. This outcome may be influenced by the extensive genetic analysis of B19V recently conducted in China that also has a high prevalence of the 1a subtype [21,22]. 

The 2023 outbreak sequences clustered together; however, B19V sequences from previous years were also part of that cluster. Three unique synonymous mutations, i.e., mutations that do not affect the predicted protein sequence, characterized this cluster. The 2023 outbreak was not geographically localized, as B19V sequences were identified in individuals residing in different regions of the country. Moreover, clinical cases of B19V infections that were not localized to specific regions were identified by physicians across the country, albeit without any laboratory confirmation. It is noteworthy that Israel, spanning only 20,770 km^2^, experiences continuous interaction across its regions. Therefore, it is not feasible to expect a correlation between specific regions and different isolates in such a widespread outbreak. Together, these findings suggest that a B19V 1a strain that has been continuously circulating in Israel also characterized the 2023 outbreak.

One plausible explanation for the rise in B19V infections in 2023 could be the lingering impact of the COVID-19 pandemic, which has significantly affected global health systems and public health priorities. Indirect factors related to COVID-19, such as changes in healthcare utilization and altered immune responses, could have influenced B19V circulation, similar to the effects observed on the prevalence of other pathogens [23,24,25]. A lower incidence of B19V infections during the pandemic years was reported in Dutch blood donors [26]. The COVID-19 pandemic’s impact on the transmission of various pathogens has been observed worldwide [27]. Israel’s distinctive response to the pandemic may have influenced the spread of viral infections in the country. Israel was among the first nations to commence COVID-19 vaccinations, emphasizing the importance of prompt action in outbreak management. Early in the pandemic, the Ministry of Health (MoH) in Israel implemented rigorous containment measures, including multiple lockdowns, stringent travel restrictions, widespread testing, self-isolation protocols, and mandatory mask-wearing [28]. Consequently, despite the early arrival of COVID-19 in February 2020, Israel has been widely commended for its effective pandemic response. Moreover, we have noted a decline in PCR-positive B19V cases during 2021–2022, potentially influenced by the measures implemented during the COVID-19 pandemic.

In the period from 2010 to 2022, we observed a recurring pattern of B19V infection where the majority of cases occurred during the early spring/summer months. The most recent significant outbreak was in 2016. Indeed, B19V epidemics were reported to occur at intervals of several years [4]. Our results suggest that the outbreak pattern occurring every 5–6 years for B19V has been maintained, and a marginal increase in the prevalence of B19V PCR-positive cases was also observed in 2011 and in 2019. The seasonal pattern was also preserved, with a notable increase in B19V cases during the early spring/summer followed by a decline in the last three months of 2023. In July 2023, approximately 50% of the tested samples were laboratory-confirmed. A study conducted in Ireland spanning from 1996 to 2008 also identified annual cycles of B19V peaking in late winter/spring, resembling the findings reported herein. Furthermore, a six-year cycle for B19V outbreaks was observed in that study [9]. Our results suggest that the outbreak cycle every 5–6 years of B19V has been maintained, despite a dramatic increase in the number of B19V infections observed in 2023. The seasonality was also preserved, with a notable increase in B19V cases in the early spring/summer followed by a decline in the last three months of 2023. In July 2023, approximately 50% of the tested samples were laboratory-confirmed. 

When comparing the demographics of B19V-positive cases in 2023 to previous years, we observed a similar pattern. The majority of requests for diagnosis and infections were observed in children and in women between the ages of 20 and 39, which corresponds to the child-bearing ages. This pattern can be explained by the fact that children and women of fertile age are the primary population groups suspected of B19V illness or pregnancy complications, thus leading to more frequent testing in these groups. Similarly, in Ireland, the majority of diagnostic tests for presumptive B19V infection in 1996–2008 were conducted in children and women between the ages of 20 and 39, corresponding the child-bearing age [9].

An additional conclusion that indirectly arises from the current study is that during significant outbreaks, cases with unexplained rash and fever, like those that test negative for measles and rubella, should undergo laboratory confirmation and should also be tested for B19V. Indeed, in a previous study in Bulgaria, 22% of measles- and rubella-negative samples collected between 2004 and 2013 were B19V positives [29]. During 2023, we detected B19V infection in samples from children initially suspected to have measles, but who tested negative for this virus and were subsequently referred for B19V molecular testing.

A group that may require special attention, particularly during a B19V outbreak, is women of child-bearing age, especially those in the early stages of pregnancy. B19V vertical transmission, which occurs in 33–51% of cases of maternal infection, can lead to feta infection and increase the risk of abortion due to hydrops fetalis or result in fetal anemia [30]. If diagnosed promptly, fetal anemia can be managed with intrauterine transfusion, with perinatal high survival rates ranging from 67% to 85% [31]. Therefore, B19V screening during pregnancy should be considered, especially in times of such an outbreak.

One limitation of this study is the lack of comprehensive data on the overall number of B19V infections in Israel in 2023 and in previous years. Similar to many European countries, B19V infection is not a notifiable disease in Israel, and epidemiological data are sparsely collected. Furthermore, B19V RT-PCR-based diagnosis is not routinely performed in clinical laboratories, and reports describing molecular detection of this virus from other countries are also sporadic and scarce. Additionally, complete B19V genome sequences are not routinely collected in Israel or in other countries, resulting in limited genomic information. However, a recent study that assessed reports of B19V infection using medical records of 2.7 million individuals in the Maccabi Healthcare Services in Israel, also identified the 2023 outbreak. This study found that most infections were diagnosed in school-aged children and pregnant women. In that study, more than 40% of the total number of infections identified during the nine investigated years (2015–2023) were documented in 2023. The study also concluded that the adjusted incidence rate ratio (IRR) of B19V in 2023 was 6.6 (95% CI 6.33–6.89) when compared to previous years, 2015 to 2022 [11].

In summary, our study illuminated the significant B19V outbreak that occurred in 2023, likely originating from a circulating 1a genotype strain, aiming to draw the attention of public health services to B19V infections and their potential outcomes. Increased awareness and understanding of B19V epidemiology and its impact can ultimately aid in the development of effective prevention and control measures to mitigate the spread of the virus and minimize associated morbidity and mortality.

## Figures and Tables

**Figure 1 viruses-16-00480-f001:**
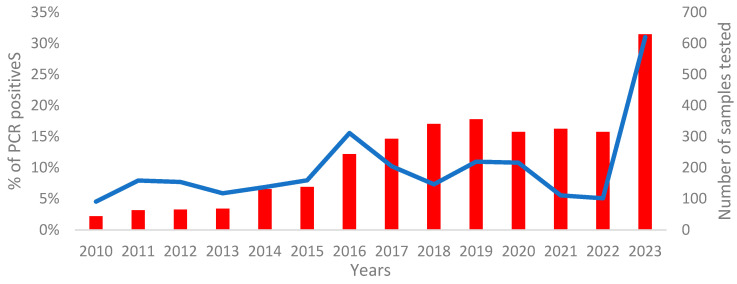
Number of samples and percentage of B19V PCR positive cases, 2010–2023. The blue line presents the % of PCR-positive samples and the red bar the number of samples tested.

**Figure 2 viruses-16-00480-f002:**
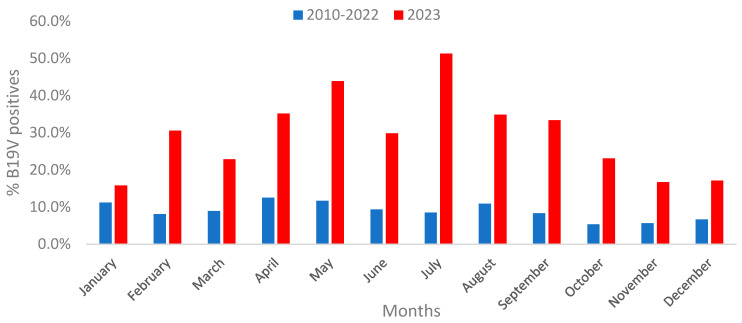
Percentage of B19V-positive PCR diagnoses per month, 2010–2022 and 2023. The blue bars present the % of B19V positives in 2010–2022 and the red bars the % of B19V positives in 2023.

**Figure 3 viruses-16-00480-f003:**
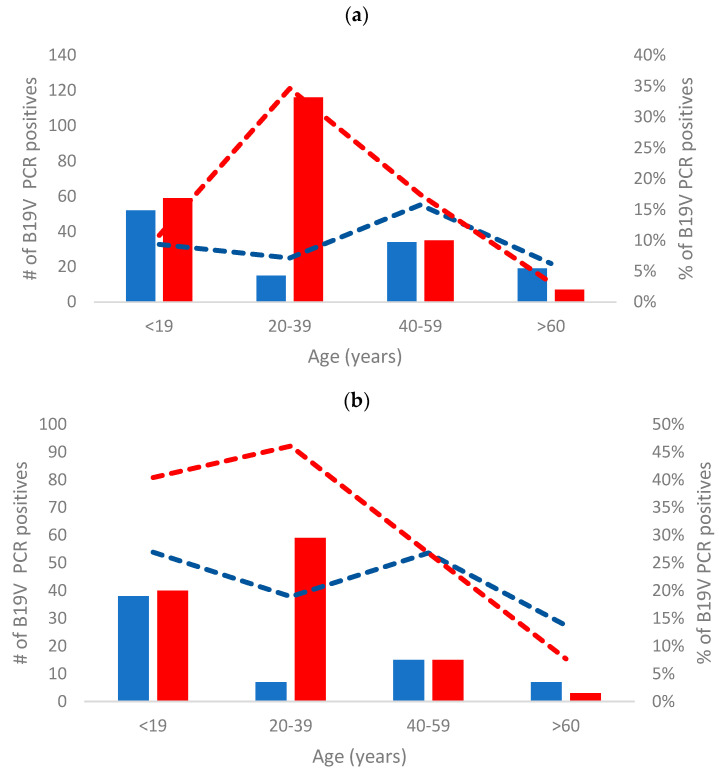
B19V PCR-positive diagnosis categorized by gender and age in 2010–2022 (**a**) and in 2023 (**b**). The blue bars present males, red bars present females. The blue and red dashed lines present the % of B19V PCR-positive male (blue lines) and female (red lines) samples in the different age groups.

**Figure 4 viruses-16-00480-f004:**
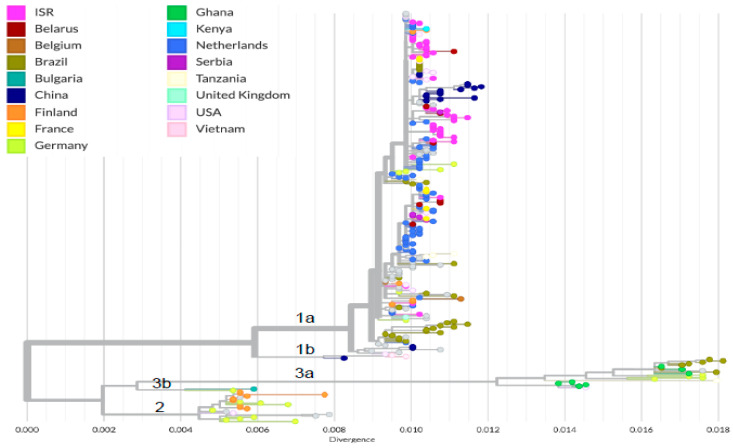
Phylogenetic tree of global (*n* = 246) and Israeli B19V (*n*= 44) sequences. The colors denote the country of origin for each sequenced sample. The B19V genotypes (1a/b, 2 and 3a/b) are also indicated.

**Figure 5 viruses-16-00480-f005:**
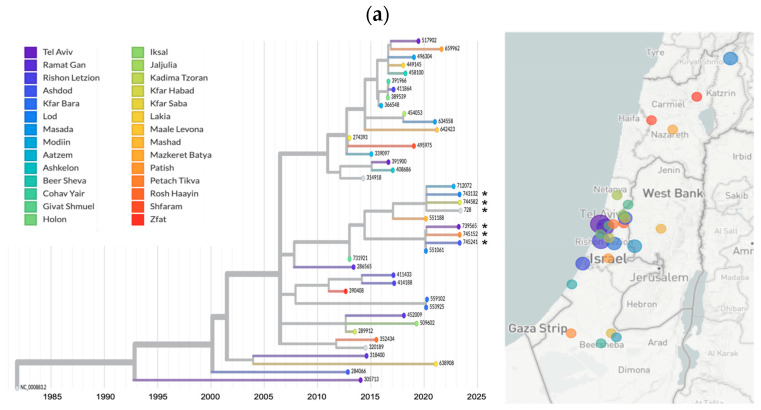
Molecular epidemiology of B19V in Israel. (**a**) Phylogenetic tree of the B19V Israeli sequences (*n* = 44) and a map of Israel with the location of each sequenced patient sample. The colors denote the patients’ location in Israel. The root is the B19V reference sequence (NC_00083.2). Asterisks indicate the 2023 cases. (**b**) Phylogenetic divergence tree presenting the genomic differences. The colors denote the year of B19V diagnosis. The dashed square highlights the cluster harboring the 2023 cases (purple). The unique mutations of this cluster are shown.

## Data Availability

All data supporting reported results can be obtained from the contributing authors upon written request.

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
