# Peer review of "Parvovirus B19 Outbreak in Israel: Retrospective Molecular Analysis from 2010 to 2023"

_viruses, 2024, doi:10.3390/v16030480_

Round 1

Reviewer 1 Report

Comments and Suggestions for Authors

This study documents the outbreak of Parvo virus B19 in Israel in 2023, which can be considered in conjunction with the results of a long-term study from 2010 to 2023, as well as the time of spread of the corona in 2021-2022 and the epidemic in 2023, and is worthy of publication in this journal. However, the following points require improvement in the analysis.

1 Introduction, It should be stated in the introduction that the route of infection of Barbovirus B 19 is through respiratory tract infection.

2 line 58 to 59: Where it describes the PT PCR procedure, reference should be made to the RT PCR method.

3 line68 to 71: The sequenced 1 kbase PCR fragment should be mentioned whether it was examined as a product of RT-PCR or whether the genome was re-amplified.

4: The phylogenetic tree generated by the GTR model should be compared with other methods such as MEGA.

5 LINE 77 to 78: The raw data of the sequences examined should be added to the supplement.

6 LINE 80: Python scripts should be included in the supplement.

7 For Figure 1, a footnote should be added to explain the meaning of line and bar graphs.

8 Figures 2 and 3 should be captioned in the same way.

9 The results of the analysis in Fig. 4 should not be limited to stating that the Israeli strain of Parvovirus is 1 A, but the regional proximity and commonality of strains should also be mentioned. In particular, it is necessary to describe the prevalent strains in regions where there is a large regional gap with other countries, such as China, whereas European countries have frequent interactions with each other as the European Union.

10 For Figure 5a, only a short statement that there is no correlation between region and isolates is mentioned. However, the data in Figure 5 should be analyzed in conjunction with more detailed information, such as the frequency of human interaction between regions. For example, an analysis should be added to correlate the evolutionary distance based on the GTR model between the two isolates with the distance between cities.

11 Similarly for Figure 5b, the year of collection should be correlated with the number of nucleotide substitutions or evolutionary distance

12 LINE 190-192 The relationship of COVID19 to the pandemic should be analyzed with particular interest in Israel, because Israel is the first country in the world to have a pandemic of COVID19. This is because Israel was the first country in the world to vaccinate against COVID19, which should have relatively controlled COVID19 transmission. It is highly possible that the wearing of masks during that period was different from that in other countries. Therefore, a more detailed comparison with the Netherlands and other countries should be made to determine what differences there might have been.

13   LINE 193 to 197: Long-term trends from 2010 to 2023 should be included in the survey of infected parvovirus 19B cases. Discussion should include whether there is a significant change in PCR positivity rates over a 5-year cycle, whereas the number of survey samples shows a long-term increasing trend.

14 LINE 240 to 245: The authors cite a literature reference of a modified infection rate of 6.6%. We request additional explanation of the relationship between this figure and the 30% infection rate shown in Figure 1 so that the reader can understand it.

15. The authors should indicate why the sampling numbers in 2023 are higher than in other years

Author Response

Reviewer 1

We thank the reviewer for the comments; please see our answers below in red.

Comments and Suggestions for Authors

This study documents the outbreak of Parvo virus B19 in Israel in 2023, which can be considered in conjunction with the results of a long-term study from 2010 to 2023, as well as the time of spread of the corona in 2021-2022 and the epidemic in 2023, and is worthy of publication in this journal. However, the following points require improvement in the analysis.

1 Introduction, It should be stated in the introduction that the route of infection of Barbovirus B 19 is through respiratory tract infection.

Reply: We amended the introduction as follows:” Infection of the respiratory tract with B19V can be asymptomatic or lead to a spectrum of clinical presentations.”

2 line 58 to 59: Where it describes the PT PCR procedure, reference should be made to the RT PCR method.

Reply: Corrected, reference added

3 line68 to 71: The sequenced 1 kbase PCR fragment should be mentioned whether it was examined as a product of RT-PCR or whether the genome was re-amplified.

Reply: The sequencing was performed on a different PCR product. This product was made using the same sample used for the initial RT-PCR diagnosis. We revised this sentence as follows: “For sequencing purposes, a 1.1 kb PCR fragment spanning the NS1/VP1U region of the virus was generated, utilizing the remains of the nucleic acids extracts used for the initial RT-PCR diagnosis and primers targeting nucleotides 1863-2953 of the NS1/VP1u B19V region”.

4: The phylogenetic tree generated by the GTR model should be compared with other methods such as MEGA.

Reply: In the manuscript we utilized the Nextstrain phylogenetic analysis pipeline, which includes the IQ-Tree program under the generalized time reversible (GTR) model to generate the tree and TreeTime for a time-resolved tree, as we also incorporated the date of collection for each sample. GTR is the most general neutral, independent, finite-sites, time-reversible model possible, and is usually used to estimate viral evolution [Sohpal et al, 2020 doi: 10.5808/GI.2020.18.3.e30; Young et al, 2022 doi: 10.3390/v14040774]. We have been using this model for the analysis with many viruses, including HIV [Zuckerman et al., doi: 10.1097/QAD.0000000000002057], dengue [Zuckerman et al., doi: 10.3390/v15122334], poliovirus [Zuckerman et al., doi: 10.2807/1560-7917.ES.2022.27.37.2200694] and more. MEGA is a suite of pipelines that can also be used to construct phylogenetic trees and has an option to apply the GTR model as well, which is the model we would have chosen if we had used the MEGA program to conduct this analysis instead of the Nexstrain pipeline.

5 LINE 77 to 78: The raw data of the sequences examined should be added to the supplement.

Reply: We have now added these data to the supplementary material. “Sequences and data describing the sequenced samples is provided in supplementary Tables 1 and 2”.

6 LINE 80: Python scripts should be included in the supplement.

Reply: Thank you, we ended up utilizing SAMtools suite to calculate the coverage and depth of sequencing and not a custom script and have now changed this in the manuscript – “Coverage and depth of sequencing were calculated using the SAMtools suite [16]”

7 For Figure 1, a footnote should be added to explain the meaning of line and bar graphs.

Reply: Thank you. We corrected the footnote as follows “Figure 1: Number of samples and percentage of B19V PCR positive cases, 2010-2023. The blue line presents the % of PCR positive samples and the red bar the number of samples tested”.

8 Figures 2 and 3 should be captioned in the same way.

Reply: Thank you. The revised caption for fig 2 and 3 (figure 3 was also revised as suggested by reviewer 2) is as follows:

Figure 2: Percentage of B19V positive PCR diagnoses per months, 2010-2022 and 2023. The blue bars present the % of B19V positives in 2010-2022 and the red bars the % of B19V positives in 2023.

Figure 3: B19V PCR-positive diagnosis categorized by gender and age in 2010-2022 (3a) and in 2023 (3b). The blue bars present males, red bars present females. The blue and red dashed lines present the % of B19V PCR positive male (blue lines) and female (red lines) samples in the different age groups.

9 The results of the analysis in Fig. 4 should not be limited to stating that the Israeli strain of Parvovirus is 1 A, but the regional proximity and commonality of strains should also be mentioned. In particular, it is necessary to describe the prevalent strains in regions where there is a large regional gap with other countries, such as China, whereas European countries have frequent interactions with each other as the European Union.

Reply: Thank you for your comment. To better assess our results, we added the following paragraph to the results section: “This cluster contains also sequences from other countries, mainly countries in Europe like the Netherlands, France and Belarus and also a tight cluster of 1a sequences from China”. 

We have also added the following sentence to the discussion section: ” Unlike other viruses, sequencing of B19V is rarely performed and therefore global data on B19V transmissions is lacking. Nonetheless, our analysis reveals that B19V sequences from Israel exhibit clustering with other 1a sequences originating from neighboring European countries. This clustering phenomenon may be attributed to the close interactions among these countries, facilitated by factors such as membership in the European Union and ongoing connections with Israel. Notably, sequences from China classified as 1a formed a distinct cluster within this Israeli group. This outcome may be influenced by the extensive genetic analysis of B19V recently conducted in China that also has a high prevalence of the 1a subtype (https://doi.org/10.1038/s41598-023-43158-y and https://doi.org/10.1016/j.onehlt.2023.100602).”.

10 For Figure 5a, only a short statement that there is no correlation between region and isolates is mentioned. However, the data in Figure 5 should be analyzed in conjunction with more detailed information, such as the frequency of human interaction between regions. For example, an analysis should be added to correlate the evolutionary distance based on the GTR model between the two isolates with the distance between cities.

Reply: To better describe the data presented in figure 5a and the situation observed in our study, we amended the discussion as follows: “Moreover, clinical cases of B19V infections which were not localized to specific regions were identified by physicians across the country, albeit without any laboratory confirmation. It is noteworthy that Israel, spanning only 20,770 km2, experiences continuous interaction across its regions. Therefore, it is not feasible to expect a correlation between specific regions and different isolates in such a widespread outbreak”.

Regarding correlation between the number of mutations and the distance between the locations in the 2023 outbreak – it is a good idea, however all the cases that were sequenced in 2023 are from cities in central Israel that are localized within a very short distance from one another (Tel-Aviv, Lod, Petach-Tikva, etc.). Due to this short distance, there’s a continuous transition of individuals across this area rendering such an analysis non-informative.

11 Similarly for Figure 5b, the year of collection should be correlated with the number of nucleotide substitutions or evolutionary distance

Reply: Thank you for this suggestion, however since the samples represent different cases from different transmission chains, some of which are probably imported from other countries, this phylogenetic tree does not represent a single transmission chain and therefore we do not necessarily expect a clear correlation between the evolutionary distance and the number of substitutions. We did perform this analysis (see attached figure below), and it shows a trend of an increase in the number of mutations during time, however since this isn’t a single transmission chain and does not provide additional insights into the scope of this manuscript, we chose to not include this in the manuscript.

12 LINE 190-192 The relationship of COVID19 to the pandemic should be analyzed with particular interest in Israel, because Israel is the first country in the world to have a pandemic of COVID19. This is because Israel was the first country in the world to vaccinate against COVID19, which should have relatively controlled COVID19 transmission. It is highly possible that the wearing of masks during that period was different from that in other countries. Therefore, a more detailed comparison with the Netherlands and other countries should be made to determine what differences there might have been.

Reply: Thanks for this important comment. To better assess the issue of the COVID-19 pandemic effect, we have revised the discussion as follows: “The COVID-19 pandemic's impact on the transmission of various pathogens has been observed worldwide (10.1002/jmv.28401). Israel's distinctive response to the pandemic may have influenced the spread of viral infections in the country. Israel was among the first nations to commence COVID-19 vaccinations, emphasizing the importance of prompt action in outbreak management. Early in the pandemic, the Ministry of Health (MoH) in Israel implemented rigorous containment measures, including multiple lockdowns, stringent travel restrictions, widespread testing, self-isolation protocols, and mandatory mask-wearing (10.3201/eid2609.201476). Consequently, despite the early arrival of COVID-19 in February 2020, Israel has been widely commended for its effective pandemic response. Moreover, we have noted a decline in PCR-positive B19V cases during 2021-2022, potentially influenced by the measures implemented during the COVID-19 pandemic”.

13   LINE 193 to 197: Long-term trends from 2010 to 2023 should be included in the survey of infected parvovirus 19B cases. Discussion should include whether there is a significant change in PCR positivity rates over a 5-year cycle, whereas the number of survey samples shows a long-term increasing trend.

Reply: Thank you for this comment. We included in the discussion the following paragraph: “Our results suggest that the outbreak pattern occurring every 5-6 years for B19V has been maintained, and a marginal increase in the prevalence of B19V PCR positive cases was also observed in 2011 and in 2019”.

14 LINE 240 to 245: The authors cite a literature reference of a modified infection rate of 6.6%. We request additional explanation of the relationship between this figure and the 30% infection rate shown in Figure 1 so that the reader can understand it.

Reply: Thanks for this comment. To clarify this point, we revised as follows: “In that study, more than 40% of the total number of infections identified during the nine investigated years (2015-2023) were documented in 2023. The study also concluded that the adjusted incidence rate ratio (IRR) of B19V in 2023 was 6.6 (95% CI 6.33–6.89) when compared to previous years, 2015 to 2022 [11].

  1. The authors should indicate why the sampling numbers in 2023 are higher than in other years

Reply: The sampling numbers in 2023 are higher because of the outbreak that was observed by the clinicians. In the introduction we have described it as follows:” In early 2023, pediatric physicians across Israel reported a notable rise in cases clinically diagnosed as the fifth disease. Simultaneously, the CVL and other clinical laboratories experienced a surge in requests for B19V diagnosis [11], although, still, most B19V cases remained clinically diagnosed without laboratory confirmation.”

Reviewer 2 Report

Comments and Suggestions for Authors

The retrospective analyses of B19 Parvovirus outbreaks in Israel between 2020 and 2023 is an interesting study to provide a better understanding about the epidemiology of this pathogen. The manuscript is clearly structured, the data in general well presented and the conclusions carefully drawn. However, there are some minor issues the authors should address prior to publication:

1)     Figure 2 would significantly benefit from additional information regarding the predominant B10 positive diagnosis observed in women of age 20-39. Thus, the overall number of patients (m vs. f) tested in the different groups remains elusive and there is no information regarding potential disease association and/or prognostic screening procedures in order to prevent miscarriage.

2)     Fig. 1: Annotations blue line % of PCR positive/red bars number of samples tested?

3)     Figure 4: The resolution is not good enough to identify individual countries in the diagram. 

Author Response

Reviewer 2: Comments and Suggestions for Authors

We thank the reviewer for the comments, please see our answers below in red.

The retrospective analyses of B19 Parvovirus outbreaks in Israel between 2020 and 2023 is an interesting study to provide a better understanding about the epidemiology of this pathogen. The manuscript is clearly structured, the data in general well presented and the conclusions carefully drawn. However, there are some minor issues the authors should address prior to publication:

  • Figure 2 would significantly benefit from additional information regarding the predominant B10 positive diagnosis observed in women of age 20-39. Thus, the overall number of patients (m vs. f) tested in the different groups remains elusive and there is no information regarding potential disease association and/or prognostic screening procedures in order to prevent miscarriage.

Reply: Thank you for your comment. To show the number of positive cases from all tested male and female samples we changed the graphs to also display the percentage of male and female positive cases by age groups (see revised figure 3). Regarding the clinical diagnosis of all parvo B19 cases and especially of those in women 20-39, as a laboratory that provides the molecular diagnosis only, we unfortunately do not have such data. However, based on our results we suggest to screen pregnant women, especially during outbreak.

  • 1: Annotations blue line % of PCR positive/red bars number of samples tested?

Reply: Thank you. We corrected the footnote to describe the annotations as follows “Figure 1: Number of samples and percentage of B19V PCR positive cases, 2010-2023. The blue line presents the % of PCR positive samples and the red bar the number of samples tested”

3)     Figure 4: The resolution is not good enough to identify individual countries in the diagram.

Reply: thank you for this comment, we have now revised figure 4 to reflect divergence instead of a timeline, which improves the resolution and separation between the different branches of the countries.

Round 2

Reviewer 1 Report

Comments and Suggestions for Authors

The paper has been appropriately revised. The dynamics of virus transmission in Israel are important in considering the spread of COVID-19 virus vaccines in high-incidence settings. The authors have made appropriate changes and added data and should be published as is.